# Assessing Barriers and Utilization of Sexual and Reproductive Health Services among Female Migrant Workers in Vietnam

**DOI:** 10.3390/ijerph20146368

**Published:** 2023-07-15

**Authors:** Toan Ha, David Givens, Hui Shi, Trang Nguyen, Nam Nguyen, Roman Shrestha, Linda Frank, Stephen L. Schensul

**Affiliations:** 1Department of Infectious Diseases and Microbiology, School of Public Health, University of Pittsburgh, Pittsburgh, PA 15261, USA; dlg43@pitt.edu (D.G.); hus38@pitt.edu (H.S.); frankie@pittt.edu (L.F.); 2Institute of Social and Medical Studies, Hanoi 10000, Vietnam; nttrang@isms.org.vn (T.N.); ntnam@isms.org.vn (N.N.); 3Department of Allied Health Sciences, University of Connecticut, Storrs, CT 06269, USA; roman.shrestha@uconn.edu; 4Department of Public Health Sciences, University of Connecticut School of Medicine, Farmington, CT 06030, USA; schensul@uchc.edu

**Keywords:** sexual and reproductive health, contraception, migrant women workers, industrial zone, Vietnam

## Abstract

Young migrant women workers frequently experience disparities in accessing health services, including sexual and reproductive health (SRH) services, especially in urban settings. This study assesses the barriers and utilization of SRH services and explores factors associated with the utilization of these services among young female migrant workers working in the industrial zone (IZ) in Vietnam. A cross-sectional survey was conducted among 1061 young women migrant workers working in an IZ in Hanoi, Vietnam. Multivariable logistic regression analysis was used to identify factors associated with utilization of SRH services. Nearly 35% of the participants reported using SRH services at least once since working in the IZ. Additionally, around 78% of the participants reported using a contraceptive method during their last sexual encounter. The study also found that older participants (25–29 years old) were nearly two times more likely to use SRH services than younger participants (18–24 years old) (OR = 1.91, 95% CI: 1.19–3.06). Married participants had nearly six times higher odds of using SRH services compared to single participants (OR = 5.98, 95% CI: 3.71–9.63), and participants with higher incomes were more likely to use SRH services (OR = 1.02, 95% CI: 1.01–1.04). The most commonly reported barriers to access SRH services were inconvenient hours of service operation (26.2%), followed by long distance from the service location (9.2%) and high service cost (5.2%). This study found a low level of SRH service utilization and identified several barriers to accessing these services among the study participants. The study findings provide important evidence insights for policymakers and program managers to develop and implement policies that help reduce barriers and enhance the provision of SRH services tailored to the needs of IZ married and unmarried women migrant workers in the IZ in rapidly developing and urbanizing countries like Vietnam and other low- and middle-income countries with similar contexts.

## 1. Introduction

Vietnam has emerged as one of the fastest growing economies globally in recent years, thanks to its adoption of the Doi Moi “open-door” policy and liberalized economic model in 1986 [1]. The foreign investment and rapid industrialization that quickly followed contributed to a steady influx of young people from rural areas seeking job opportunities and better wages in the country’s major urban centers. To accommodate this ongoing and rapid industrial expansion, the government of Vietnam established specially designed industrial zones (IZs). As of 2021, Vietnam had a total of 284 operational IZs with approximately 3.78 million workers, of which the great majority were women [2].

In Vietnam, the young women who work in the IZs come from low-income patriarchal families, where they typically have had limited educational opportunities and social mobilities [3]. IZ jobs represent one of the only opportunities for most rural women to obtain regular wages to support themselves and their families. However, the IZ environment offers limited personal growth, education, social independence, and autonomy. Gender inequality, patriarchal norms, workplace structures, and economic constraints contribute to health disparities among these women. Consequently, their abilities to negotiate safer sex and exercise control over their sexual activities are often compromised, exposing them to various sexual health risks [4,5]. The social isolation of life away from home, the possibility for promotion at work and reinforcement of high-risk sexual behaviors by peer groups and intimate partners, and the lack of access to accurate sexual health information increases risks such as unplanned pregnancies, intimate partner violence, and sexually transmitted infections [6].

Gender disparities in workplace violence and partner violence against women are of great concern. A 2019 review by Santoro et al. highlighted the prevalence of gender-based violence in the workplace, including physical harassment and sexual abuse, which disproportionately affect women compared to men [7]. Among the female factory workers, in Bangladesh, a study conducted among factory women workers revealed common instances of violence against women in the workplace, such as physical and verbal abuse, constant pressure, personal restrictions, and withheld pay [8]. In China, another study reported that 70 percent of female factory workers experienced some form of gender-based violence and harassment at work, resulting in 15 percent quitting their jobs [9]. These findings underscored the vulnerabilities faced by women migrant workers in factory environments.

Studies in Vietnam indicate that women migrants are more likely to have received lower levels of education than male peers, lack access to sexual and reproductive health information, and report low levels of condom use by sexual partners [10]. A recent study among IZ women workers in Vietnam found that premarital sex among unmarried migrant worker was 12.6%, and most of these sexually active workers used contraceptives inconsistently [11]. These limited data suggest a high potential for sexual health risks and negative health outcomes among populations of IZ migrant women workers. Despite large numbers of unmarried women migrant workers working in IZs, there are no official government programs providing SRH services for them. Furthermore, in Vietnam, migrant workers who are not permanent city residents are not covered by SRH services at their current residences, which results in them having to pay high out-of-pocket payments when seeking care [10]. Consequently, many migrant workers turn to self-treatment or seeking health care services only for extreme need.

While other recent research has focused on women migrant workers’ exposure to sexually transmitted infections and human immunodeficiency virus (HIV) [12,13], limited attention has been given to migrant workers’ utilization of sexual and reproductive health services (SRH), despite the importance of addressing their needs [14]. A systematic review of SRH among women migrant workers in the southeast Asian nations found low awareness of SRH issues and limited knowledge about sexually transmitted infections and contraception among these women [14]. Recent studies among rural to urban women migrant workers working in Chinese factories revealed higher rates of unprotected sex and lower contraceptive use [12], high levels of unmet contraceptive needs [15], and a substantial proportion reported having one or more abortions [16]. However, there is a paucity of research on the use of contraception and SRH service utilization among sexually active young women migrant workers working in IZs in Vietnam. This study aims to fill this research gap by gathering data on SRH service utilization and identifying barriers through direct engagement with workers in a marginalized IZ setting in Vietnam.

Specifically, the study’s objectives include: (1) assess the utilization and barriers to SRH services, and (2) explore the factors associated with SRH services utilization among IZ women migrant workers. The study’s findings will provide valuable insights into the existing conditions and barriers related to SRH services for women migrant workers in industrial zones (IZs). Additionally, it will shed light on the unmet needs experienced by these migrant women workers. This knowledge will be crucial in informing the development of policies and targeted interventions aimed at reducing barriers and improving access to SRH services among IZ migrant women workers in Vietnam, which may be also relevant other low- and middle-income countries with similar social cultural settings.

## 2. Materials and Methods

### 2.1. Study Design and Sample Participants

This study utilized data from the study concerning “HIV risk among young women migrant workers in the industrial zones in Vietnam”, which is described in detail in prior publications [17]. A cross-sectional study was conducted among young women migrant workers between January and November 2020 in Thang Long Industrial Park. The main objective of the original study was to examine the prevalence of HIV and associated risks among female migrant workers who were either single or currently married but not living with a husband or partner, as it was hypothesized that this specific group was more likely to engage in risky sexual behavior [13]. Subsequently, we utilized the gathered dataset to evaluate the barriers and utilization of sexual and reproductive health services within this population.

Eligibility criteria for participation in the study involved self-identifying as (1) a woman; (2) being between 18 and 29 years of age; (3) being either single, currently married but not living with a husband or partner while working in the IZ, separated, divorced, or widowed; (4) having worked in the IZ for six or more months; and (5) being from a rural area or another province prior to IZ employment

### 2.2. Sample Size and Recruitment Procedures

We employed cluster sampling to select participants, which involved choosing groups of women who lived or worked in close proximity, such as those in rent clusters or dormitories. To account for the potential similarities in behaviors among participants within the same cluster, we utilized intra-class correlation within the primary sampling units to calculate the effective sample size [18]. As a result, we determined that a total of 1061 participants were needed for the survey.

A multistage, clustered sampling method was used to select the participants [18]. First, 779 rent clusters and two dormitories located near the IZ were mapped and quantitatively assessed for suitability. For study feasibility, only rent clusters with six or more eligible participants were selected, resulting in 419 qualifying rent clusters and 360 excluded rent clusters. Once eligible rent clusters and dormitories were compiled, women in the eligible age group were identified by dormitory/property managers and contacted directly for screening and participation by the research team and local collaborators who were women and former IZ workers. Within the 419 rent clusters, a total of 1316 eligible migrant workers were eligible to participate in the study; however, 320 of them declined to participate. This resulted in a total of 936 participants living in rent clusters being selected for interviews. Of the total of 175 eligible residents from the two selected dormitories, 125 participants agreed to participate in the interviews. Thus, the final total number of young female workers interviewed from both rent clusters and dormitories was 1061.

### 2.3. Data Collection

The research team organized an after-hours meeting for eligible women migrants at a nearby community health center. The women were given complete information about the study, including the activities they were expected to participate in, the topics covered by the survey, the time required for the interview, their right to refuse or withdraw, and the measures to ensure confidentiality of the interviews. Participants were then presented with written informed consent to confirm that they agreed to join the interview. Each participant took part in a face-to-face anonymous interview using structured questionnaires conducted by trained field researchers in a private setting. The interview lasted for an hour. All interviewers received training in interviewing techniques, developing rapport, ensuring confidentiality, and answering questions raised by participants.

This study was reviewed and approved by the Institutional Review Boards of the University of Connecticut Health Center, USA, and the Institute of Social and Medical Studies, Hanoi, Vietnam (IRB approval number: 19-134O-1, date of ethical approval: 18 April 2019).

### 2.4. Measurements

#### 2.4.1. Sociodemographic Characteristics

The demographic characteristics of participants, including age, marital status, education, ethnicity, income, housing status, and working hours, were assessed.

#### 2.4.2. Use of SRH Services

The use of SRH services was assessed through self-report using a “yes” or “no” response to a single question about whether the individual had utilized any of the specific SRH services offered by private or government health facilities since starting work in the industrial zone. A positive (“yes”) response to any one of these services was regarded as service utilization. If participants responded positively with a “yes”, further validation was conducted by asking additional questions about the type of SRH services they had utilized. If they provided a positive response to at least one of these services, it was considered as service utilization, categorizing the individual as having utilized SRH services. In the regression analysis, the utilization of SRH services was operationalized as a binary variable, where a “yes” response indicated service utilization, and a “no” response indicated non-utilization.

#### 2.4.3. Use of Contraceptives

Participants were asked if either themselves or their partner used any method to avoid a pregnancy during the last time that they had sexual intercourse (yes/no). If the participant answered “yes”, they would be further asked “What contraceptive method did you or your partner use?” If the participants answered “No”, they would instead be further asked “Why did not you use any contraceptives the last time you had sex?”

#### 2.4.4. Barriers to SRH Services

Participants were asked to identify any barriers they encountered when accessing SRH services from a pre-determined list. This list included options such as SRH clinic inconvenient hours, negative/unwelcoming attitudes of staff, stigma and discrimination concerns, expensive services, inconvenient location(s), lack of confidentiality, and an open response for any other barriers that participants had experienced.

### 2.5. Statistical Analysis

Descriptive statistics were used to describe characteristics of the study sample using frequencies, means, and standard deviations (SD). Differences in use of contraceptive methods and barriers to SRH services between participants were examined using Chi-squared or Fisher’s exact tests. Bivariable binary logistic analysis was conducted to select potential variables for the multivariable logistic regression of factors associated with utilization of SRH services. Variables with *p*-values less than 0.25 in bivariable logistic regression model were included into the multivariable binary logistic regression model. Nagelkerke R^2^ and χ^2^ were used to explain the model and effect size. A *p*-value was considered significant at <0.05. Effect estimates were reported using an odds ratio with a corresponding 95% confidence interval. All analyses were performed using Stata 17.0 (Stata Corp, College Station, TX, USA).

## 3. Results

### 3.1. Characteristics of Participants

A total of 1061 young women migrant workers were recruited and interviewed. The mean age was 23.2 years (range: 18–29 years old, SD = 3.3) (Table 1). Over two thirds of the respondents (*n* = 794) were single. A majority of respondents had completed high school (79.4%) and lived in rent clusters (88.2%). Mean income was USD 294 per month, ranging from USD 195 to USD 652 (SD = 56.5). Compared to single participants, married participants in the study were more likely to be older (*p* < 0.001), more likely to have worked more years in the IZ (*p* < 0.001), and have a higher monthly income (*p* < 0.05). A greater proportion of single respondents lived in rent clusters than married participants (*p* < 0.05).

### 3.2. Sexual Activities and Use of Contraceptive Methods

A total of 512 participants (215 were single and 297 were married) reported ever having sexual intercourse, as shown in Table 2. Nearly 78% of the participants who reported engaging in sexual activity reported using a contraceptive method to prevent pregnancy during their last sexual encounter. The most commonly reported contraceptive method was condoms (76.4%), followed by withdrawal (23.6%), daily contraceptive pills (10.5%), intrauterine devices (IUD) (8.8%), and other methods, including “safe days” (the calendar method) and emergency contraception. Single sexually active participants were more likely to use a contraceptive method during their last sexual intercourse compared to married participants (85.6% vs. 72.1%; *p* < 0.001).

The top reasons cited for not using contraceptives during the last sexual encounter were not feeling the need (34.2%), not liking to use them (16.7%), and other factors such as wanting to conceive or already being pregnant (21.1%) (Figure 1). Regarding decision-making on use of contraception, most women in both groups indicated that they and their partner(s) jointly made the final decision on contraceptive use (*n* = 310, 77.9%).

### 3.3. Barriers to SRH Services

The most commonly reported barriers to accessing SRH services were inconvenient clinic hours (reported by 25.5% of participants, *n* = 270), followed by the distance from the service location (8%, *n* = 85) and the high cost of services (5.2%, *n* = 55) (Table 3). Single women migrant participants were more likely than married participants to report that the distant location of SRH services was a barrier (*p* < 0.05). The cost of the service was cited as a major barrier by more married participants than single participants, but the difference was not statistically significant (*p* > 0.05).

### 3.4. Factors Associated with Utilization of SRH Services

About one third (35%) of the participants utilized SRH services since working in the IZ. When broken down by marital status, the utilization rate was higher among married participants (71.4%) compared to unmarried participants (20.1%), as seen in Table 3. Table 4 presents the multivariable logistic regression analysis results on factors associated with the utilization of SRH services. The overall model was found to be statistically significant (χ^2^(15) = 145.60; *p* < 0.001), and the model explained around 21% of the variance in the utilization of SRH services (Nagelkerke R2 = 0.205). The results showed that participants aged 25–29 were nearly twice as likely to use SRH services compared to younger (18–24 years old) participants (OR = 1.91, 95% CI: 1.19–3.06). Married participants had nearly six times higher odds of using SRH services compared to unmarried participants (OR = 5.98, 95% CI: 3.71–9.63). Additionally, participants with higher incomes were more likely to use SRH services (OR = 1.02, 95% CI: 1.01–1.04).

## 4. Discussion

This study evaluated the utilization of SRH services among female migrant workers in Hanoi, Vietnam, and identified factors related to their service utilizations. The results showed a low utilization rate of SRH services among the participants, with only 35% reporting usage since working in the IZ. These findings underscored the necessity for expanded access to SRH services and targeted efforts to enhance utilization among young migrant workers in the IZ in Vietnam.

One major finding of this work is highlighting the stark difference among married and unmarried women workers in this IZ. As noted in the results section above, unmarried participants (10.7%) were dramatically less likely to use SRH services compared to married participants (72.2%). Similar findings have been reported among female migrant workers working in factories in China, where unmarried migrant workers reported a much lower rate of accessing SRH services than married workers in that setting as well [19]. The observed differences in SRH service utilization among unmarried young women in our study could have been due to several factors within the cultural context of Vietnam. Firstly, SRH programs for unmarried young people are generally not prioritized in the country, leading to limited availability of services specifically tailored for this group. Secondly, there is a persistence of negative attitudes and stigma surrounding premarital sexuality in Vietnam, which creates barriers for unmarried sexually active women to access SRH services, including discomfort and embarrassment [20]. These results indicate that additional SRH service programs and interventions most need to prioritize unmarried migrant young women workers in Vietnam.

The study found that 77.7% of participants reported using a contraceptive method during their last sexual encounter to prevent pregnancy. Notably, the most commonly used contraceptive method among the participants was male condoms, which aligned with previous research conducted among unmarried female migrant workers in China and Vietnam [11,21]. This finding suggested that actively promoting condom use among industrial workers, especially among unmarried participants, would likely be an effective and accepted intervention. This preference for condoms may have been attributed to their dual function of providing contraception and protection against sexually transmitted infections (STIs). Given that our study population primarily consisted of unmarried women, it was plausible that the desire to prevent both unintended pregnancies and the risk of STIs could have been driving the high usage of condoms. Therefore, when advocating for a campaign promoting condoms as an effective and cost-effective contraceptive method, it was important to emphasize the dual protection capabilities of condoms in preventing both pregnancy prevention and STI risk reduction.

One of the major barriers to accessing SRH services that participants reported was “location is too far”. Addressing the challenge of distant location for migrant workers to access SRH services is crucial for ensuring equitable access. A number of feasible solutions could be implemented to bridge the geographical distance and enhancing accessibility to vital SRH care for migrant workers, which include establishing mobile SRH clinics that travel to areas where migrant workers reside and training private and family physicians to deliver basic SRH services for migrant workers within the local communities. Moreover, leveraging mHealth (which utilizes mobile technologies to enhance access to health information and services) and virtual platforms could play a crucial role in addressing the geographical barrier. By utilizing technologies such as virtual consultations and mHealth apps, migrant workers could remotely access SRH services, receive professional advice, and access relevant health information. Recent studies among young people and other populations have indicated that social media and mHealth app interventions have the potential to improve maternal health, mental health, and knowledge about pregnancy [22,23].

Other common barriers to accessing SRH services included cost and inconvenient hours of service operation, which were consistent with previous findings among migrant workers in Asia [14,24,25] and other low- and middle-income countries [26]. For instance, a study of migrant workers in China and Malaysia showed that cost was a significant barrier to accessing healthcare services [27]. To facilitate the utilization of SRH services among migrant workers, advocating for the coverage of SRH services for migrant workers through government-mandated migrant health insurance programs and/or the provision of subsidies for accessing SRH services could reduce financial burdens and ensure essential SRH care for this vulnerable population. In addition, the development of effective collaborations between the IZ employees and local health care centers could help the service’s delivery. For example, collaborative interventions such as support for local health care centers extending hours of available services or employers creating flexibility in working conditions/hours so that migrant workers can use the services when needed.

There were limitations to our study. First, the study participants were recruited from one industrial zone, which may not have been representative of all industrial zones in Vietnam. Therefore, caution should be exercised when generalizing these findings to other rural-to-urban women migrant workers or other industrial zones. The study findings should be interpreted only as associations, as the study employed a cross-sectional design and did not demonstrate causality. Additionally, pre-marital sex is still highly stigmatized in Vietnam and may be underreported by participants in our study. This social barrier needs to be further explored through qualitative research. Similarly, the use of SRH services was self-reported and not confirmed by medical records, which may introduce recall bias and social desirability bias. It is worth mentioning that the study did not inquire about the frequency of contraceptive use. Therefore, the fact that participants used contraception or not during the most recent encounter may not have been representative of their typical contraceptive use overall. The lack of information about behavioral patterns of self-reported contraceptive use could seriously skew results and limit conclusions. Nevertheless, these limitations were generally outweighed by the study’s strengths. The study boasted a large sample size among a highly marginalized, under-resourced socio-economic group and examined critical topics that could enhance their health and wellbeing—namely, multiple facets surrounding SRH service utilization and barriers to SRH services among single and married women migrant workers.

## 5. Conclusions

The study provides evidence that IZ migrant workers currently encounter several barriers related to sexual and reproductive health services and information, which previous research has linked to various health risks and unplanned pregnancies. The study’s findings have significant policy and program implications, which may also be applicable in other low- and middle-income countries with similar socio-cultural settings. Specifically, there is need to develop and implement targeted policies that prioritize the provision of SRH services tailored to the needs of married and unmarried IZ women migrant workers. Moreover, efforts should be made to strengthen coordination and collaboration between government agencies, IZ employers, non-governmental organizations, and healthcare providers. This collaboration should focus on providing subsidy services and reducing barriers to SRH services for migrant workers. Finally, it is crucial to implement comprehensive sexuality education outreach programs within the IZs to ensure that young migrant workers have access to accurate information about SRH. Implementing these policy and program measures may help overcome barriers, enhance access to services, and improve SRH outcomes for IZ women migrant workers.

## Figures and Tables

**Figure 1 ijerph-20-06368-f001:**
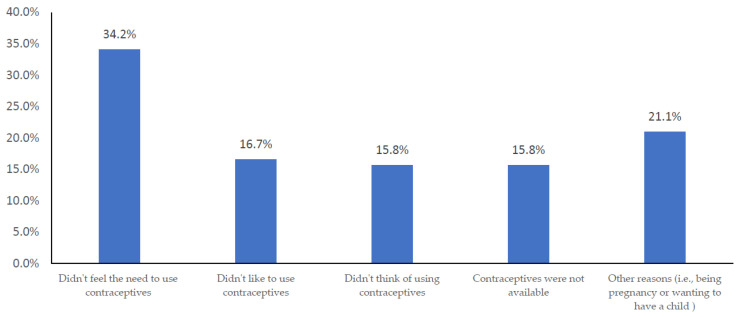
Reasons for nonuse of contraceptive methods at the last sexual encounter (*n* = 512).

**Table 1 ijerph-20-06368-t001:** Sample characteristics among participants (*n* = 1061).

Variables	Mean (SD) or *n* (%)
Age (years), mean (SD)	23.2 (3.3)
Education, *n* (%)	
Under high school	109 (10.3)
High school	842 (79.4)
Above high school	110 (10.4)
Ethnicity, *n* (%)	
Kinh (majority)	735 (69.3)
Other	326 (30.7)
Residence, *n* (%)	
Rent cluster	936 (88.1)
Dormitory	125 (11.8)
Years working in IZ, *n* (%)	
≤1 year	483 (51.7)
2–5 years	429 (31.4)
>5 years	149 (17.2)
Working hours per day, *n* (%)	810 (76.3)
8 h	251 (23.7)
>8 h	
Monthly income (USD), mean (SD)	291.9 (56.5)

Note: Continuous variables are presented as mean (SD); categorical variables are presented as proportions, *n* (%).

**Table 2 ijerph-20-06368-t002:** Contraception use among young women migrant workers.

Characteristics	Total *n* = 1061 (%)	Single *n* = 755 ^a^ (74.8)	Married *n* = 264 ^b^ (25.2)	*p*
Ever had sexual intercourse				<0.001
Yes	512 (50.3)	215 (29.8)	297 (100.0)	
No	507 (49.7)	507 (70.2)	0 (0.0)	
Used any contraceptive method to avoid a pregnancy during last sexual intercourse ^b^				<0.001
Yes	398 (77.7)	184 (85.6)	214 (72.1)	
No	114 (22.3)	31 (14.4)	83 (27.9)	
Ever had an abortion				<0.001
Yes	55 (10.7)	13 (5.2)	42 (15.9)	
No	457 (89.3)	235 (94.8)	222 (84.1)	
Place where had an abortion				0.51
Private clinic	27 (41.7)	7 (53.8)	20 (47.6)	
Private hospital	5 (11.1)	1 (7.7)	4 (9.5)	
Commune health center	6 (11.1)	0 (0.0)	6 (14.3)	
Public hospital/health Center	17 (36.1)	5 (38.5)	12 (28.6)	
Used the SRH/HIV service while living in the IZ				<0.001
Yes	368 (34.7)	153 (20.1)	215 (71.4)	
No	693 (65.3)	607 (79.9)	86 (28.6)	
The most common contraceptives methods				
Daily pill				<0.001 **
Yes	42 (10.5)	7 (3.8)	35 (16.4)	
No	356 (89.5)	177 (96.2)	179 (83.6)	
Intrauterine device (IUD)				<0.001 **
Yes	35 (8.8)	0 (0.0)	35 (16.4)	
No	363 (91.2)	184 (100.0)	179 (83.6)	
Condom (male condom)				<0.001 **
Yes	304 (76.4)	167 (90.8)	137 (64.0)	
No	94 (23.6)	17 (9.2)	77 (36.0)	
Withdrawal				0.183
Yes	94 (23.6)	29 (15.8)	24 (11.2)	
No	345 (86.7)	155 (84.2)	190 (88.8)	
Other contraceptive methods				0.089
Yes	13 (3.3)	3 (1.6)	10 (4.7)	
No	385 (96.7)	181(98.4)	204 (95.3)	
Person suggested the use of contraceptive methods.				0.014 *
Partner	12 (3.0)	7 (3.8)	5(2.3)	
Herself	76 (19.1)	24 (13.0)	52 (24.3)	
Joint decision	310 (77.9)	153 (83.2)	157 (73.4)	

Note: ^a^ 39 cases did not respond to the question ever having sex; ^b^ among those who reported having sex; IZ: industrial zone; IUD: Intrauterine device; * *p* < 0.05; ** *p* < 0.001.

**Table 3 ijerph-20-06368-t003:** Barriers to SRH services among young women migrant workers.

Barriers to SRH Services	Total	Single	Married	*p*
	*n* = 1061 *n* (%)	*n* = 794 (74.8%)	*n* = 267 (25.2%)	
Timing is not convenient.				0.33
Yes	270 (25.5)	208 (26.2)	62 (23.2)	
No	791 (74.5)	586 (73.8)	205 (75.8)	
Quality of services is not good.				0.10
Yes	13 (1.2)	12 (1.5)	1 (0.37)	
No	1048 (98.8)	782 (98.5)	266 (99.6)	
Expensive				0.31
Yes	55 (5.2)	38 (4.8)	17 (6.4)	
No	1006 (94.8)	756 (95.2)	250 (93.6)	
Discrimination				0.31
Yes	7 (0.66)	6 (0.77)	1 (0.37)	
No	1054 (99.3)	788 (99.2)	266 (99.6)	
Lack of confidentiality				0.20
Yes	13 (1.2)	11 (1.4)	2 (0.75)	
No	1048 (98.8)	783 (98.6)	265 (99.3)	
The location is too far				0.01 *
Yes	85 (8.0)	73 (9.2)	12 (4.5)	
No	976 (92.0)	721 (90.8)	255 (95.5)	

Note: SRH: sexual and reproductive health; * *p* < 0.05.

**Table 4 ijerph-20-06368-t004:** Factors associated with usage of the SRH services among migrant women workers.

Characteristics	AOR	95% CI	*p*
Age (years)			
18–24	1 (ref)	—	
25–29	1.91	1.19–3.06	0.007 *
Marital status, *n* (%)			
Unmarried	1 (ref)	—	
Married	5.98	3.71–9.63	<0.001 **
Ethnicity			
Other minority groups	1 (ref)	—	
Kinh	1.37	0.89–2.12	0.157
Education background			
5–9 years	1 (ref)	—	
10–12 years	1.46	0.82–2.61	0.201
≥12 years	1.42	0.65–3.09	0.374
Monthly income (USD)	1.02	1.01–1.04	0.005 *
Residence type			
Dormitory	1 (ref)	—	
Rent cluster	1.65	0.89–3.03	0.109
Knowledge of HIV/AIDS	1.10	0.89–1.35	0.362
Use of any contraceptive to prevent a pregnancy at last sex			
No	1 (ref)	—	
Yes	0.69	0.42–1.14	0.146
Number of health problem	1.01	0.96–1.06	0.729
Depressive symptoms			
No	1 (ref)	—	
Yes	1.06	0.43–2.62	0.906
HIV stigma			
Low	1 (ref)	—	
High	1.16	0.70– 1.90	0.567
Perceived HIV risk			
Low	1 (ref)	—	
Median	0.88	0.57–1.37	0.579
High	0.92	0.36–2.31	0.856
Ever having taken HIV test			
No	1 (ref)	—	
Yes	1.34	0.82–2.17	0.240

Note: The overall multivariable model statistical summary: χ^2^(15) = 145.60; *p* < 0.001, Nagelkerke R2 = 0.2054. AOR: adjusted odds ratio; CI: Confidence Interval; SRH/HIV: sexual and reproductive health/human immunodeficiency virus; ** *p* < 0.001; * *p* < 0.05

## Data Availability

The data presented in this study are available on request from the corresponding author.

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
