# Peer review of "Assessing Barriers and Utilization of Sexual and Reproductive Health Services among Female Migrant Workers in Vietnam"

_ijerph, 2023, doi:10.3390/ijerph20146368_

Round 1

Reviewer 1 Report

Congrats for your nice work!

I only suggested you to make a more focused introduction because it is too long and tiring to read. 

Also on study design bring more of your scenery about reproductive health services (coverage, accessibility, if the services are free of charge,.... ).

English need to be revised 

Author Response

Dear Reviewer,

Thank you for taking the time to review our paper. We appreciate your thoughtful and constructive reviews and suggestions. We have made suggested changes based on your suggestions and feedback. Please find our responses below in bold for easy spotting. Changes in the manuscript appear in red font to facilitate the review.

Reviewer:

Congrats for your nice work! I only suggested you make a more focused introduction because it is too long and tiring to read. 

Response: Thank you so much. We appreciate your feedback and have revised the introduction to make it more focused and succinct.

Also on study design bring more of your scenery about reproductive health services (coverage, accessibility, if the services are free of charge,.... ).

Response: Thank you for your suggestion. We will incorporate additional information about reproductive health services in the study introduction instead of the design section, specifically focusing on aspects such as coverage, and associated costs. By including this information in the introduction, we aim to provide the audience with a comprehensive overview of the existing reproductive health services and their implications, rather than confining it to the study design section. This will help contextualize the study within the broader landscape of reproductive health services and highlight the importance of addressing gaps and barriers in access.

Reviewer 2 Report

Dear authors,

Thank you for the opportunity to review your manuscript “Assessing barriers and utilization of sexual and reproductive health services among female migrant workers in Vietnam”. This study is very interesting and well written, and the results were particularly well thought out, with tables to keep the reader’s attention and interesting correlation analyses performed. However, I have a few concerns I’d like you to address:

1.      Layout needs to be revised, there are many double spaces between words thorough the manuscript.

2.      In the introduction, I would advise to analyse more in depth the gender differences (briefly mention on line 64) in workplace violence and partner violence (just as an example you can read: Santoro PE et al. “Occupational hazards and gender differences: a narrative review”).

3.      In the introduction, on line 129, this sentence is not clear: “but may be used to help inform improvements”. What did you mean by this? Could you reformulate?

4.      In the Methods section you mentioned you used data from the study “HIV risk among young women migrant workers in the industrial zones in Vietnam.” Was this approved by an ethics committee? Do you have the ID number? The ethics approval needs to be stated in the manuscript and the ID number needs to be added.

5.      In the results, a significant barrier to SRH services access is “location is too far”.  Unless I misread something, this is not discussed (lines 332-335), but it is very interesting: what can we do to overcome this barrier? Make SRH services available (at least somewhat) even to family physicians? Make transportation more available? As the only statistically significant barrier I believe this should be addressed.

6.      Similarly, you discuss condoms being the most used method of contraception: unlike IUDs and birth control pills, this is also protective for STDs, do you believe this could be the reason (especially given the population you’re studying of mostly unmarried women)? Should this be addressed in the discussion when you suggest to campaign for condoms specifically as an effective and cost-effective contraceptive method (lines 312-315)?

This study is very interesting and I believe it should be accepted for publication after the aforementioned revisions are made.

Author Response

Dear Reviewer,

Thank you for taking the time to review our paper. We appreciate your thoughtful and constructive reviews and suggestions. We have made suggested changes based on your suggestions and feedback. Please find our responses below in bold for easy spotting. Changes in the manuscript appear in red font to facilitate the review.

Reviewer:

Dear authors,

Thank you for the opportunity to review your manuscript “Assessing barriers and utilization of sexual and reproductive health services among female migrant workers in Vietnam”. This study is very interesting and well written, and the results were particularly well thought out, with tables to keep the reader’s attention and interesting correlation analyses performed. However, I have a few concerns I’d like you to address:

1. Layout needs to be revised, there are many double spaces between words thorough the manuscript.

 Response: Thank you. We have revised the layout. All double spaces have been removed and replaced with single spaces. The entire introduction and layout have also received additional edits for clarity and succinctness, which are highlighted in red.

2. In the introduction, I would advise to analyse more in depth the gender differences (briefly mention on line 64) in workplace violence and partner violence (just as an example you can read: Santoro PE et al. “Occupational hazards and gender differences: a narrative review”).

Response: Thank you for your valuable suggestion. We have now included an additional paragraph in the paper to provide a more in-depth analysis of gender differences in workplace violence and partner violence, in particular in the context of women working in the factory. Lines 71-80

3.In the introduction, on line 129, this sentence is not clear: “but may be used to help inform improvements”. What did you mean by this? Could you reformulate?

Response: We have revised the paragraph to provide greater clarity.  Lines 113-119

4. In the Methods section you mentioned you used data from the study “HIV risk among young women migrant workers in the industrial zones in Vietnam.” Was this approved by an ethics committee? Do you have the ID number? The ethics approval needs to be stated in the manuscript and the ID number needs to be added.

Response: We have added the ethics approval statement, IRB number, and date of ethical approval in the paper: IRB number: 19-134O-1, date of ethical approval: 10/13/2020. Line 176.

5. In the results, a significant barrier to SRH services access is “location is too far”.  Unless I misread something, this is not discussed (lines 332-335), but it is very interesting: what can we do to overcome this barrier? Make SRH services available (at least somewhat) even to family physicians? Make transportation more available? As the only statistically significant barrier I believe this should be addressed.

Response: Thank you for your excellent suggestion. We have now included a paragraph discussing solutions addressing this significant barrier in the discussion. Lines 340-353

6. Similarly, you discuss condoms being the most used method of contraception: unlike IUDs and birth control pills, this is also protective for STDs, do you believe this could be the reason (especially given the population you’re studying of mostly unmarried women)? Should this be addressed in the discussion when you suggest campaigning for condoms specifically as an effective and cost-effective contraceptive method (lines 312-315)?

Response: Thank you for your valuable feedback. We have incorporated a discussion on the dual protection capabilities of condoms, highlighting condom effectiveness in preventing both unintended pregnancies and reducing the risk of sexually transmitted infections. Lines 331-338

Reviewer 3 Report

The paper is good, however, there are some concerns which should be addressed before final decision.

1. Please use shorter paragraphs in your introduction and discussion. Long paragraphs are unwanted.

2. Ethical concerns: Please provide number and date of your ethical approval.

3. Provide supplementary materials with all sociodemographic questions of your study. For example, provide question for the "Use of SRH services" etc.

4. Table 1: Mean and SD should be provided in appropriate places, but not in the "n" and "%" columns.

5. Lines 231-232: "reported at least one sexual activity". This should be clarified.

6. Line 235: Abbreviations should be introduced after first applying. IUD was not deciphered. Please use abbreviations correctly. In general, there are a lot of unnecessary abbreviations.

7. How was utilization of SRH services operationalized in regression analysis? This should be clarified.

8. Please provide specific practical implications of your study.

Author Response

Dear Reviewer,

Thank you for taking the time to review our paper. We appreciate your thoughtful and constructive reviews and suggestions. We have made suggested changes based on your suggestions and feedback. Please find our responses below in bold for easy spotting. Changes in the manuscript appear in red font to facilitate the review.

Reviewer:

The paper is good, however, there are some concerns which should be addressed before final decision.

1. Please use shorter paragraphs in your introduction and discussion. Long paragraphs are unwanted.

Response: Thank you. We have now shortened the introduction and discussion. Specifically, we reduced the length of each paragraph making it easy to read and follow.

2. Ethical concerns: Please provide the number and date of your ethical approval.

Response: we have now added the following IRB Number and date of ethical approval in the paper: IRB number: 19-134O-1, date of ethical approval: 10/13/2020. Line 176

3. Provide supplementary materials with all sociodemographic questions of your study. For example, provide question for the "Use of SRH services" etc.

Response: We have included supplementary materials containing sociodemographic questions and questions related to the "Use of SRH services" and other relevant topics in the study.

4. Table 1: Mean and SD should be provided in appropriate places, but not in the "n" and "%" columns.

Response: Thank you for pointing this out. We have carefully placed Mean and SD in appropriate places in Table 1.

5. Lines 231-232: "reported at least one sexual activity". This should be clarified.

Response: Thank you. To clarify the meaning, we have revised the text by replacing "reported at least one sexual activity" by "reported ever having sexual intercourse." Line 257-258

6. Line 235: Abbreviations should be introduced after first applying. IUD was not deciphered. Please use abbreviations correctly. In general, there are a lot of unnecessary abbreviations.

Response: We have reviewed the text and ensure that all abbreviations are properly introduced after first applying including Intrauterine device (IUD) (line 262). In addition, we have also reduced the number of unnecessary abbreviations in the paper.

7. How was utilization of SRH services operationalized in regression analysis? This should be clarified.

 Response:  In the regression analysis, the utilization of SRH services was operationalized as a binary variable. We have now added the following paragraph to the paper to provide a clear understanding of how the utilization of SRH services was operationalized in our regression analysis: “Participants were asked a single question about whether they had utilized any specific SRH services offered by private or government health facilities since starting work in the industrial zone. A "yes" response indicated service utilization, while a "no" response indicated non-utilization. To validate the utilization, participants who responded positively were asked additional questions about the type of SRH services they had utilized. If they reported utilizing at least one of these services, it was considered as service utilization, categorizing them as having utilized SRH services. In the regression analysis, this utilization of SRH services was then treated as a binary variable, with "yes" indicating utilization and "no“ indicating non-utilization. Line 187-194.

Pls see the supplementary materials for further details of this question.

8. Please provide specific practical implications of your study.

Response: Thank you for your valuable feedback. We have now included specific practical implications of the study in both policy and program development as well as service delivery. Line 390-404

Round 2

Reviewer 2 Report

The manuscript can be accepted for publication. 

English language is okay

Author Response

Dear Reviewer,

Thank you for taking the time to review our paper again and recommend it for publication. We greatly appreciate your thorough review and positive evaluation of our work. Our response is below.

Reviewer: The manuscript can be accepted for publication. 

Author response: Thank you.

Reviewer 3 Report

The paper was improved. This is a laconic manuscript which in my opinion can be accepted for publication after some minor changes.

Please provide descriptions of all abbreviations in your tables.

What does "aOR" mean in Table 4?

See also various sizes of font in the notes of your tables. This should be unified. 

Author Response

Dear Reviewer,

Thank you for taking the time to review our paper again and recommend it for publication. We greatly appreciate your thorough review and positive evaluation of our paper. We have made suggested changes based on your feedback.  Our response is below.

1. Reviewer: The paper was improved. This is a laconic manuscript which in my opinion can be accepted for publication after some minor changes.

Author response: Thank you. We appreciate your valuable feedback. We have made these minor changes as you have suggested.

2. Reviewer: Please provide descriptions of all abbreviations in your tables.

Author response: We have included a list of abbreviations in the note section for all Tables.

3. Reviewer: What does "aOR" mean in Table 4?

Author response: We apologize for not being clear. "aOR" stands for Adjusted Odds Ratio (AOR). We have replaced “aOR” with “AOR” and included a note at the end of the Table clarifying its meaning as Adjusted Odds Ratio.

 4. Reviewer: See also various sizes of font in the notes of your tables. This should be unified. 

Author response: Thank you. We have unified the sizes of the fonts in the notes of all tables as per your suggestion.
